# MUC1 Expressions and Its Prognostic Values in US Gastric Cancer Patients

**DOI:** 10.3390/cancers15040998

**Published:** 2023-02-04

**Authors:** Young-Il Kim, Robert Luke Pecha, Tara Keihanian, Michael Mercado, S. Valeria Pena-Munoz, Kailash Lang, George Van Buren, Sadhna Dhingra, Mohamed O. Othman

**Affiliations:** 1Section of Gastroenterology and Hepatology, Department of Medicine, Baylor College of Medicine, Houston, TX 77030, USA; 2Center for Gastric Cancer, National Cancer Center, Goyang 10408, Republic of Korea; 3Department of Pathology and Immunology, Baylor College of Medicine, Houston, TX 77030, USA; 4Michael E. DeBakey Department of Surgery, Baylor College of Medicine, Houston, TX 77030, USA

**Keywords:** gastric cancer, mucin, prognosis

## Abstract

**Simple Summary:**

Although the alteration of mucin (MUC) expression was associated with gastric cancer (GC) prognosis, its prognostic values remain to be elucidated in US GC patients. The positive expression of MUC1 was associated with aggressive pathologic features of GC including diffuse-type cancer, advanced cancer, lymph node metastasis, and distant metastasis. In early GC, patients with a high level of MUC1 expression showed a higher rate of lymphovascular invasion, which is a strong risk factor for lymph node metastasis and noncurative resection after endoscopic submucosal dissection. MUC1 expression can be a useful prognostic marker in US GC patients.

**Abstract:**

This study aims to evaluate the prognostic value of MUC expression in US GC patients. A total of 70 tumor specimens were collected from GC patients who underwent surgery or endoscopic resection between 2013 and 2019 at a tertiary referral center in the US. MUC expression status including MUC1, MUC2, MUC5AC, and MUC6 was evaluated by immunohistochemical staining. The positive rates of MUC1, MUC2, MUC5AC, and MUC6 were 71.4%, 78.6%, 74.3%, and 33.3%, respectively. Patients with positive MUC1 expression had a significantly higher rate of aggressive pathologic features including diffuse-type cancer (42.0% vs. 0%; *p* < 0.001), advanced GC (80.0% vs. 30.0%, *p* < 0.001), lymph node metastasis (62.0% vs. 20.0%; *p* = 0.001), and distant metastasis (32.0% vs. 5.0%; *p* = 0.017) compared with those with negative MUC1 expression. However, the differences in the pathologic features were not observed according to MUC2, MUC5AC, and MUC6 expression status. In early gastric cancer (EGC), patients with a high level of MUC1 expression showed a higher rate of lymphovascular invasion (71.4% vs. 21.4%; *p* = 0.026) and EGC meeting non-curative resection (85.7% vs. 42.9%; *p* = 0.061) than those with negative MUC1. In US GC patients, MUC1 expression is associated with aggressive pathological features, and might be a useful prognostic marker.

## 1. Introduction

Globally, gastric cancer is one of the most common cancers, and it is the fifth most commonly diagnosed cancer and the fourth leading cause of death according to the Global Cancer Statistics 2020 [1]. Unlike the global statistics, gastric cancer is a relatively rare cancer in the US and represented 1.4% of all new cancer cases in 2022 [2]. However, the prognosis of gastric cancer still remains poor in the US, and approximately two-thirds of new gastric cancers (62%) were diagnosed with regional (25%) or distant (37%) stages whose 5-year survival rates were 32.9% and 5.9%, respectively [2]. Because a better prognosis was shown in the localized stage (5-year survival rate, 71.8%), early detection of cancer is important to improve gastric cancer prognosis.

Studies investigating the molecular features of gastric cancer have proposed several genomic subtypes of gastric cancers according to gene expression data [3,4]. However, these molecular classifications proposed are limited to use clinically. Because gastric cancer is a heterogeneous cancer morphologically, biologically, and genetically [5], the clinical significance of the genetic subtypes is not well established. The biomarkers recommended in the current clinical practice guideline include human epidermal growth factor receptor 2 (HER2), microsatellite instability (MSI) status, and programmed death ligand 1 (PD-L1) [6] These biomarkers are focused on predicting drug response in patients with advanced gastric cancer who receive chemotherapy. Thus, there has been great importance in finding a biomarker for the early detection and prognosis of gastric cancer.

Mucins (MUCs) are high molecular weight glycoproteins expressed throughout the gastrointestinal tract [7]. Among 21 MUC genes, the main MUCs in the stomach include membrane-bound mucins (MUC1) and secreted mucins (MUC5AC and MUC6) [8]. Among the mucins, MUC1 is normally expressed in the glandular or epithelial cells of the breast, gastrointestinal tract (esophagus, stomach, duodenum, and pancreas), uterus, prostate, and lungs [9]. Experimental studies have suggested that MUC1 functions as an oncoprotein and plays an important role in multiple steps of carcinogenesis including tumor proliferation, metabolism, invasion, and metastasis [10,11], and MUC1 overexpression was associated with poor outcomes in lung cancer and breast cancers [12,13]. Likely, studies have reported significant associations between the positive expression of MUC1 and clinicopathological features with poor prognosis of gastric cancer, including the advanced clinical stage [14,15] and positive lymph node metastasis [15,16]. Although the findings suggest the role of MUC1 as a prognostic marker for gastric cancer, the expression and prognostic value of MUC1 have not been studied in US patients with gastric cancer.

The present study aimed to evaluate the prognostic value of MUC1 expression status in US gastric cancer patients.

## 2. Materials and Methods

### 2.1. Patients

We reviewed, retrospectively, a pathology electronic database and selected eligible gastric cancer patients who underwent surgery or endoscopic submucosal dissection (ESD) between 2013 and 2019 at the Baylor St. Lukes Medical Center, Houston, Texas. Of these, patients who had pathological specimens available were included. Baseline demographics (age, sex, smoking, and alcohol drinking) and clinical demographics (comorbid illnesses, *Helicobacter pylori* infection status, and treatment received) were collected. The final pathological data collected were tumor size, background intestinal metaplasia status, tumor depth of invasion, Lauren classification, lymphovascular invasion, perineural invasion, lymph node metastasis, and distant metastasis.

This study was approved by the Institutional Review Board for Baylor College of Medicine and Affiliated Hospitals (Protocol Number: H-45708). Because of minimal patient risk, informed consent from patients was waived by the Institutional Review Board.

### 2.2. Gastric Cancer Treatment

Initially, gastric cancer patients underwent diagnostic workup including esophagogastroduodenoscopy with biopsy, laboratory tests, and imaging studies (chest, abdomen, and pelvis computed tomography). Patients who had a gastric cancer with potentially resectable locoregional stages received a surgical treatment. The standard radical gastrectomy with lymph node dissection was performed, and the reconstruction methods were Billorth I or II for subtotal and distal gastrectomy, esophagogastrostomy for proximal gastrectomy, and Roux-en-Y esophagojejunostomy for total gastrectomy. Open or laparoscopic surgery was selected according to patient condition and clinical tumor stages.

In patients with early gastric cancer, endoscopic resection was considered for tumors meeting the following indications: (1) adenocarcinoma, intestinal type, nonulcerated mucosal lesions ≤ 2 cm in size; (2) adenocarcinoma, intestinal type, nonulcerated mucosal lesions > 2 cm; (3) adenocarcinoma, intestinal type, mucosal lesions ≤ 3 cm with ulceration; (4) adenocarcinoma, intestinal type, early submucosal lesions ≤ 3 cm; or (5) adenocarcinoma, diffuse type, nonulcerated mucosal lesions ≤ 2 cm [17]. Curative resection was defined when all of the following conditions were fulfilled: en-bloc resection with negative resection margin (both horizontal and vertical margins), absence of lymphovascular invasion, and final pathologic findings meeting one of the above indications.

### 2.3. Immunohistochemical (IHC) Stain and Evaluation of MUC Expression

Archived gastric cancer specimens were retrieved from storage, and one representative paraffin block was selected from each case. From each block, 8 unstained formalin-fixed and paraffin-embedded sections were cut. IHC staining for MUC 1, MUC 2, MUC 5AC, and MUC6 was performed on all cases. Standard methods were used for IHC staining for MUC 1 (anti-MUC 1 antibody [ab 15481, Abcam], IgG rabbit polyclonal 1:100 dilution, 60 min incubation), MUC2 (anti-MUC 2 antibody [ab 97386, Abcam], IgG rabbit polyclonal, 1:200 dilution, 60 min incubation), MUC 5AC (anti MUC 5AC antibody [ab 218466, Abcam], IgG Mouse monoclonal, 1:250 dilution, 60 min incubation), and MUC 6 (anti MUC 6 antibody [ab 223846, Abcam], IgG rabbit monoclonal, 1:200 dilution, 60 min incubation). IHC staining of formalin-fixed, paraffin-embedded tissue sections (4 mm) was conducted using the standard streptavidin–biotin complex technique after antigen retrieval. Appropriate positive and negative controls were processed simultaneously. The IHC stained slides were reviewed by a pathologist (S.D.) for quality control and diagnostic interpretation. IHC staining was considered positive when staining was observed in ≥ 5% of the neoplastic cells. All IHC stains were independently evaluated by a pathologist (S.D.). The IHC stain was assessed as positive or negative. The positive IHC on each case then was assessed for the percentage of positive cells and scored (score 1 = 1% to 25%; score 2 = 26% to 50%; score 3 = 51% to 100%). The chromogenic signal intensity was graded from 1+ (weak) to 3+ (strong) (Figure 1). A composite score was calculated as the score of positive staining × signal intensity. The composite scores of 3 or greater were defined as high expression. Finally, the results of IHC staining were compared with clinical and pathological data.

### 2.4. Statistical Analysis

The Chi-square test or Fisher’s exact test for categorical variables and the Student *t*-test for continuous variables were used to evaluate between-group differences. The Cochrane–Armitage test or linear-by-linear association test was used to detect the trends between pathological findings and composite scores of IHC stain. All statistical analyses were performed using STATA version 17.0 (StataCorp, College Station, TX, USA). A *p*-value of less than 0.05 was considered statistically significant.

## 3. Results

### 3.1. Baseline Patient Characteristics and Pathological Findings

A total of 70 gastric cancer patients were eligible for this study, and the baseline patient demographics and clinical characteristics are shown in Table 1. Of them, 4 patients underwent ESD, and the remaining 66 patients received surgical resection. The median age of patients was 68 years, and the male proportion was 54.3%.

The mean tumor size was 3.7 cm, and more than half of the tumors (38 of the 70 patients, 54.3%) were located in the upper third of the stomach (cardia and high body) (Table 2). According to the Lauren classification, tumor histologic types were intestinal type in 46 patients (65.7%) and diffuse type in 21 patients (30.0%). The proportion of early gastric cancer (tumors within mucosa or submucosa) was 34.3% (24 of the 70 patients). Lymph node metastasis and distant metastasis were found in 35 patients (50.0%) and 17 patients (24.3%), respectively. The patients with distant metastasis were diagnosed with positive peritoneal cytology on the peritoneal washing.

### 3.2. Mucin Expression and Pathological Features of Gastric Cancer

In the resected gastric cancer tissues, mucin expression rates were 71.4% for MUC1, 78.6% for MUC2, 75.4% for MUC5AC, and 33.3% for MUC6 (Table 2). Patients with MUC1 positive expression had a higher rate of tumors with background intestinal metaplasia (*p* = 0.015), diffuse type histology (*p* < 0.001; Figure 2A,B), advanced gastric cancer (tumors invading the proper muscle or deeper; Figure 2C) (*p* < 0.001), perineural invasion (*p* = 0.027; Figure 2D–F), lymphovascular invasion (*p* = 0.015; Figure 2A,B), lymph node metastasis (*p* = 0.001), and distant metastasis (*p* = 0.017) compared to those without MUC1 expression (Table 3). The final tumor stages were more advanced in patients with MUC1 expression than in those without MUC1 expression (*p* < 0.001). However, there were no significant differences in pathological features of gastric cancer according to the expression status of MUC2, MUC5AC, and MUC6 (Appendix A).

A subgroup analysis was performed including patients who underwent surgical treatment with final tumor stage I~III on final pathological evaluations (Appendix A). The MUC1 positive patients had a significantly higher proportion of diffuse- or mixed-type histology (*p* = 0.001), tumors invading proper muscle or deeper layer (*p* = 0.002), lymph node metastasis (*p* = 0.017), and advanced final tumor stages (*p* = 0.002).

### 3.3. Pathological Features of Gastric Cancer According to MUC1 Expression Levels

The distribution of composite scores of MUC1 IHC stain was score 0 (negative) for 20 patients (28.6%), score 1–2 (low expression level) for 7 patients (10.0%), and score 3–9 (high expression level) for 43 patients (61.4%). The expression level of MUC1 was not associated with tumor histological type according to the Lauren classification (Figure 3A). As the levels of MUC1 expression increased, the pathological features showing a poor prognosis tended to increase significantly, including invasion of muscularis propria or deeper (ptrend = 0.0002; Figure 3B), perineural invasion (ptrend = 0.005; Figure 3C), lymphovascular invasion (ptrend = 0.0066; Figure 3D), lymph node metastasis (ptrend = 0.0014; Figure 3E), and distant metastasis (ptrend = 0.0094; Figure 3F).

### 3.4. MUC1 Expression in Early Gastric Cancer

Among 24 patients with early gastric cancer, the MUC1 expression was found in 10 patients (41.7%). Of the patients with MUC1 expression, seven patients had a high level of expression. The proportion of tumor histological type by the Lauren classification was not different according to the MUC1 expression level (Figure 4A). Patients with a high level of MUC1 expression showed a higher rate of lymphovascular invasion (71.4% vs. 21.4%, *p* = 0.026) compared to those without MUC1 expression (Figure 4C). The rates of submucosal invasion (85.7% vs. 42.9%, *p* = 0.061; Figure 4B) and lymph node metastasis (28.6% vs. 7.1%, *p* = 0.186; Figure 4D) were higher in patients with a high level of MUC1 expression than in those without MUC1 expression, although there was no statistical significance. According to the ESD criteria for early gastric cancer, the rate of non-curative resection was 85.7% (six of the seven patients) in patients with a high level of MUC1 expression and 42.9% (six of the fourteen patients) in those without MUC1 expression (*p* = 0.061; Figure 4E).

## 4. Discussion

In the present study, we evaluated the prognostic role of mucin expression in 70 US gastric cancer patients who underwent ESD or surgical resection. The MUC1 expression in gastric cancer tissues was significantly associated with aggressive pathological features including diffuse histologic type, deeper layer of tumor invasion, perineural invasion, lymphovascular invasion, lymph node, and distant metastasis. Moreover, there were trends of more aggressive pathological features according to the higher level of MUC1 expression. In subgroup analysis, including the early gastric cancer, the MUC1 high expression was related to the higher rate of lymphovascular invasion. To our best knowledge, this is the first report of the prognostic role of MUC1 in US gastric cancer patients.

To assess the prognosis of gastric cancer, the tumor, lymph node, and metastasis (TNM) staging is the most important parameter [18]. Our study results showed that MUC1 expression was significantly associated with pathological features of poor prognosis in gastric cancer. Patients with MUC1 expression had a higher rate of deeper tumor depth (T staging), lymph node metastasis (N staging), and distant metastasis (M staging). Thus, final tumor stages were significantly more advanced in patients with MUC 1 expression. A meta-analysis of 10 retrospective studies from Asia and Europe reported that MUC1 positive gastric cancer patients had a lower 5-year survival rate (hazard ratio 0.27, 95% confidential interval [CI] 0.11–0.66) [19]. Similar to the results of our study, MUC1 positivity had a higher rate of vascular invasion (odds ratio [OR] 1.64, 95% CI 1.13–2.39) and lymph node metastasis (OR 2.10, 95% CI 1.20–3.67) in this meta-analysis [18]. The results from previous studies and our study results suggest the role of MUC1 as a prognostic marker for gastric cancer.

Conventional serum tumor markers including carcinoembryonic antigen (CEA), carbohydrate antigen (CA)19-9, and CA-72-4 have been reported to be useful in detecting recurrence and distant metastasis, and predicting patient survival in gastric cancer [20,21]. Molecular markers such as HER2 and PD-L1 are also used to choose target agents for patients who receive combination chemotherapy because of unresectable gastric cancer [6,22,23]. Those serum and molecular markers have limited roles in early gastric cancer, and their uses are focused on patients with advanced gastric cancer treated with chemotherapy. Our study showed a possible prognostic role in early gastric cancer, and a high level of MUC1 expression by using the composite score of IHC which had more submucosal invasion and lymph node metastasis in early gastric cancer. However, there was no statistical significance mainly due to the low proportion of early gastric cancer (34.3%, 24 of the 70 patients). Further studies involving a large number of early gastric cancer patients are needed to confirm the prognostic role of MUC1 in early gastric cancer.

ESD is a minimally invasive treatment and has become a standard treatment for early gastric cancer meeting the ESD indication [6,22,23]. Although ESD is a well-established procedure in Asian countries including Japan and Korea, it has been recently adopted in the US. A large prospective multi-center study in North America reported that the curative resection rate of stomach neoplasms was 77.2% [24]. In early gastric cancer patients who revealed non-curative resection after ESD, additional surgical treatment is needed due to the risk of lymph node metastasis [22,23]. The lymphovascular invasion is the strongest risk factor for lymph node metastasis in early gastric cancer [25], and non-curative resection is confirmed if there is lymphovascular invasion on the final pathological evaluation after ESD for early gastric cancer. However, there are no diagnostic tests or markers to detect or predict lymphovascular invasion before ESD. In our study, MUC1 expression was significantly associated with the presence of lymphovascular invasion. Furthermore, subgroup analysis in early gastric cancer patients showed a significant association between the high level of MUC1 expression and lymphovascular invasion. Patients with high level of MUC1 expression had a two-fold increase in the proportion of early gastric cancer categorized as non-curative resection (85.7% vs. 42.9%) compared to those without MUC1 expression. Thus, the assessment of MUC1 on the endoscopic biopsy specimen of gastric cancer might be a useful marker when ESD is considered a primary treatment for early gastric cancer.

In a meta-analysis, MUC1 expression was associated with the intestinal-type of gastric cancer (OR 1.76; 95% CI 1.27–2.44) [19]. Unlike this result, our study showed a high rate of diffuse-type cancer in patients with MUC1 expression. These conflicting results might be due to an ethnic difference between previous studies and our study. In genome-wide association studies (GWASs), differences in frequency of the single nucleotide polymorphism (SNP) frequencies on the MUC1 gene were shown according to the Lauren classification of gastric cancer (intestinal type vs. diffuse type) and ethnic groups [26,27,28].

Although this study is the first study evaluating the prognostic roles of MUC1 expression in US gastric cancer, there are several limitations. First, this study is a retrospective study, and selection bias is not avoidable. Second, we could not perform survival analyses according to the mucin expression status because we could not receive complete survival data. However, the MUC1 expression was associated with the aggressive pathological features related to poor prognostic outcomes, including a higher rate of deeper tumor depth, lymph node, and distant metastasis. These results, in turn, suggest that MUC1 expression is a poor prognostic marker in patients with gastric cancer. Third, the proportion of early gastric cancer was not enough to provide statistical power for subgroup analysis in patients with early gastric cancer. Finally, the outcomes were not evaluated according to patient treatment methods because only four patients who underwent ESD were included. Thus, a prospective study including more early gastric cancer patients who undergo ESD will be needed to confirm our findings.

## 5. Conclusions

MUC1 is a useful prognostic biomarker for predicting gastric cancer outcomes. A high level of MUC1 expression is associated with the presence of lymphovascular invasion and a high possibility of non-curative resection after ESD in patients with early gastric cancer.

## Figures and Tables

**Figure 1 cancers-15-00998-f001:**
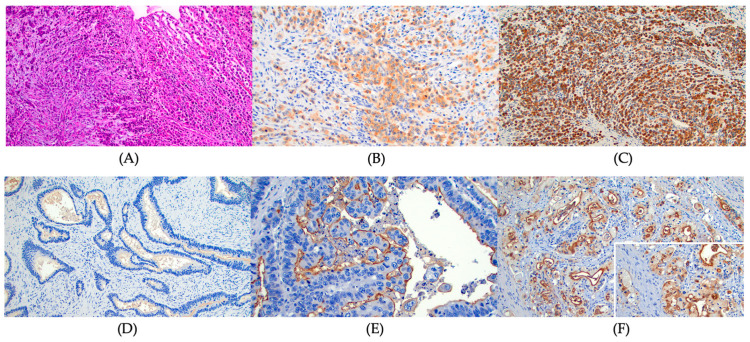
The immunohistochemical (IHC) stain for mucin expression. (**A**) Diffuse signet ring cell gastric carcinoma. Hematoxylin and Eosin stain, ×100, (**B**) MUC 1 focal cytoplasmic staining, 2+ intensity, diffuse signet ring cell gastric carcinoma. MUC 1 IHC, ×200, (**C**) MUC 1 diffuse and strong cytoplasmic staining, 3+ intensity, diffuse signet ring cell gastric carcinoma. MUC 1 IHC, ×100, (**D**) No staining of MUC1, intestinal type gastric carcinoma. MUC 1 IHC, ×200, (**E**) MUC 1 strong luminal staining only, 3+ intensity, intestinal type gastric carcinoma. MUC 1 IHC, ×200, and (**F**) MUC 1 luminal and focal cytoplasmic staining, 3+ intensity, intestinal type gastric carcinoma. MUC 1 IHC, ×200. Inset: higher magnification to show luminal and cytoplasmic staining, ×400.

**Figure 2 cancers-15-00998-f002:**
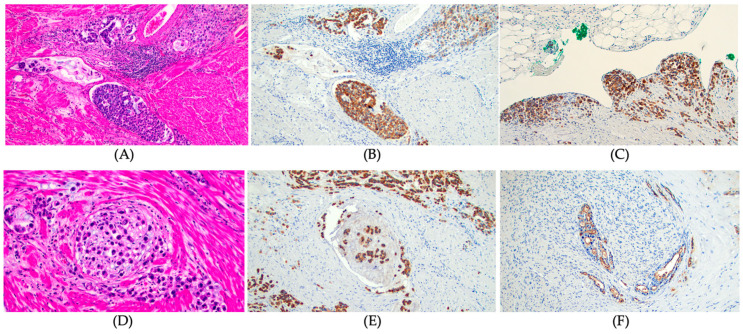
MUC1 expression and pathological features of gastric cancer. (**A**) Lymphovascular invasion, diffuse signet ring cell gastric carcinoma. Hematoxylin and Eosin stain, ×100, (**B**) Strong MUC1 expression in tumors cells invading lymphatics and vessels. MUC 1 IHC, ×100, (**C**) Strong MUC 1 expression in tumor cells infiltrating serosal surface. MUC 1 IHC, ×100, (**D**) Intraneural invasion, diffuse signet ring cell carcinoma. Hematoxylin and Eosin stain, ×200, (**E**) Strong MUC1 expression in tumor cells infiltrating nerve bundle, diffuse signet ring cell carcinoma. MUC 1 IHC, ×100, and (**F**) Strong MUC1 luminal expression in tumor cells infiltrating nerve bundle, intestinal type gastric adenocarcinoma. MUC 1 IHC, ×100.

**Figure 3 cancers-15-00998-f003:**
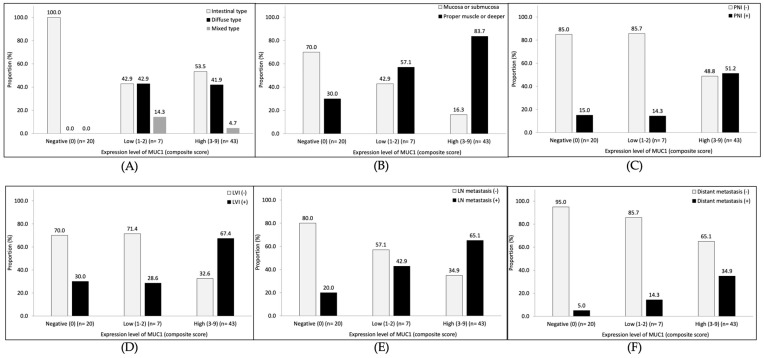
The association between the level of MUC1 expression and pathological features. (**A**) Lauren classification, (**B**) tumor depth, (**C**) perineural invasion (PNI), (**D**) lymphovascular invasion (LVI), (**E**) lymph node (LN) metastasis, and (**F**) distant metastasis.

**Figure 4 cancers-15-00998-f004:**
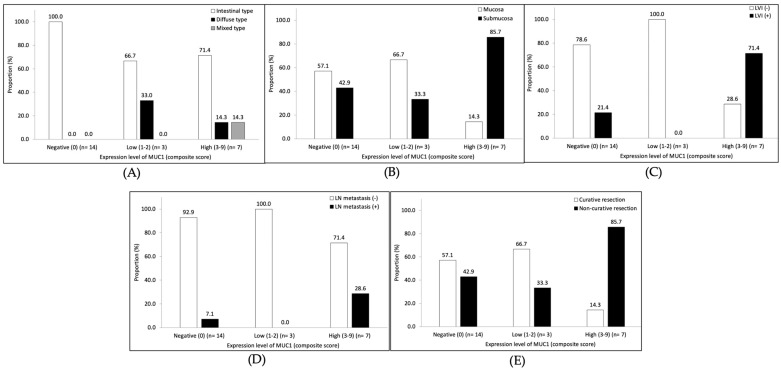
Subgroup analyses of the association between the level of MUC1 expression and pathological features in early gastric cancer patients. (**A**) Lauren classification, (**B**) tumor depth, (**C**) lymphovascular invasion (LVI), (**D**) lymph node (LN) metastasis, and (**E**) tumor category of curability according to endoscopic submucosal dissection criteria.

**Table 1 cancers-15-00998-t001:** Baseline patient demographics and clinical characteristics.

	Total No = 70
Age (year), median (IQR)	68 (58–79)
Sex, no (%)	
Female	32 (45.7)
Male	38 (54.3)
Smoking, no (%)	
Never	31 (44.3)
Former	29 (41.4)
Current	10 (14.3)
Alcohol drinking, no (%)	
Never	47 (67.1)
Former	7 (10.0)
Current	16 (22.9)
Diabetes mellitus, no (%)	21 (30.0)
*H. pylori* infection, no (%)	
Negative	36 (51.4)
Positive	9 (12.9)
Unknown	25 (35.7)
Body mass index (kg/m^2^), mean ± SD	27.5 ± 7.7
Treatment for gastric cancer, no (%)	
Endoscopic submucosal dissection	4 (5.7)
Surgery	66 (94.3)

Abbreviations: IQR, interquartile range; SD, standard deviation.

**Table 2 cancers-15-00998-t002:** Tumor characteristics and mucin expression.

	Total No = 70
Tumor size (cm), mean ± SD	3.7 ± 2.4
Tumor location, no (%)	
Upper third	38 (54.3)
Middle third	15 (21.4)
Lower third	17 (24.3)
Background intestinal metaplasia, * no (%)	
Absent	30 (42.9)
Present	40 (57.1)
Lauren classification, no (%)	
Intestinal type	46 (65.7)
Diffuse type	21 (30.0)
Mixed type	3 (4.3)
Tumor depth, no (%)	
Mucosa	11 (15.7)
Submucosa	13 (18.6)
Proper muscle or deeper	46 (65.7)
Perineural invasion, no (%)	
Absent	44 (62.9)
Present	26 (37.1)
Lymphovascular invasion, no (%)	
Absent	33 (47.1)
Present	37 (52.9)
Lymph node metastasis, no (%)	
Absent	35 (50.0)
Present	35 (50.0)
Distant metastasis, no (%)	
Absent	53 (75.7)
Present	17 (24.3)
AJCC TNM stage, ** no (%)	
Stage I	24 (34.3)
Stage II	16 (22.9)
Stage III	13 (18.6)
Stage IV	17 (32.0)
MUC expression, no/total no. (%)	
MUC1	50/70 (71.4)
MUC2	55/70 (78.6)
MUC5AC	52/69 (75.4)
MUC6	23/69 (33.3)

Abbreviations: SD, standard deviation; MUC, mucin; AJCC, American Joint Committee on Cancer. * Background intestinal metaplasia was evaluated in endoscopically or surgically resected specimen. ** Final pathological cancer stages were classified according to the 7th edition of AJCC TNM staging classification.

**Table 3 cancers-15-00998-t003:** MUC1 expression and pathological characteristics of gastric cancer.

	MUC1		*p*
	Negative	Positive	
	(No = 20)	(No = 50)	
Tumor size (cm), mean ± SD	3.1 ± 1.9	3.9 ± 2.5	0.171
Tumor location, no (%)			0.495
Upper third	5 (25.0)	12 (24.0)	
Middle third	6 (30.0)	9 (18.0)	
Lower third	9 (45.0)	29 (58.0)	
Background intestinal metaplasia, no (%)			0.015
Absent	4 (20.0)	26 (52.0)	
Present	16 (80.0)	24 (48.0)	
Lauren classification, no (%)			<0.001
Intestinal type	20 (100)	26 (52.0)	
Diffuse type	0 (0)	21 (42.0)	
Mixed type	0 (0)	3 (6.0)	
Tumor depth, no (%)			<0.001
Mucosa	8 (40.0)	3 (6.0)	
Submucosa	6 (30.0)	7 (14.0)	
Proper muscle or deeper	6 (30.0)	40 (80.0)	
Perineural invasion, no (%)			0.027
Absent	17 (85.0)	27 (54.0)	
Present	3 (15.0)	23 (46.0)	
Lymphovascular invasion, no (%)			0.015
Absent	14 (70.0)	19 (38.0)	
Present	6 (30.0)	31 (62.0)	
Lymph node metastasis, no (%)			0.001
Absent	16 (80.0)	19 (38.0)	
Present	4 (20.0)	31 (62.0)	
Distant metastasis, no (%)			0.017
Absent	19 (95.0)	34 (68.0)	
Present	1 (5.0)	16 (32.0)	
AJCC TNM stage, * no (%)			<0.001
Stage I	14 (70.0)	10 (20.0)	
Stage II	4 (20.0)	12 (24.0)	
Stage III	1 (5.0)	12 (24.0)	
Stage IV	1 (5.0)	16 (32.0)	

Abbreviations: MUC, mucin; SD, standard deviation; AJCC, American Joint Committee on Cancer. * Final pathological cancer stages were classified according to the 7th edition of AJCC TNM staging classification.

## Data Availability

The datasets generated during and/or analyzed during the current study are available from the corresponding author on reasonable request. The data are not publicly available due to patient privacy and the General Data Protection Regulation.

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
