# Peer review of "MUC1 Expressions and Its Prognostic Values in US Gastric Cancer Patients"

_cancers, 2023, doi:10.3390/cancers15040998_

Round 1

Reviewer 1 Report

To the authors:

This paper is somewhat well-written original article concerning MUC1 in gastric cancer. In this article, authors revealed that MUC1 expression might be a useful prognostic marker in US GC patientsThe objective of this study is curious and important, however, there are several points that remain unclear. 

Major Comments;

1.Details such as indications for ESD and surgical procedures should be described in the method section.

2.The authors should provide details of distant metastasis, as it is associated with lymph node metastasis; 17 cases (24.3%) had distant metastasis, however was there an indication for surgery? Patient selection criteria, if any, should also be noted.

3.Since lymph node metastasis and distant metastasis are included in the endpoints, ESD cases and surgical cases should be analyzed separately.

4.Although the author mentions in the limitation, prognostic data should also be provided if MUC1 is considered to be oncogenic.

5.“ESD non curative resection” is described, however it should be described in detail because the meaning differs depending on whether vertical or horizontal margin present.

Author Response

We submit our response as a separate word file. 

Reviewer 2 Report

In principle, this is an interesting study providing some potentially clinically useful data.

I have some minor suggestions for the Authors: 

The microscopic pictures lack scale bars, can you include them?

The tables could be formatted in order to better show statistical significance between tested parmaeters. Since some advanced methods were used by the Authors some readers e.g. clinicians may have a problem understanding the results e.g. in table 3 the parameter Tumor location is shown:

Tumor location, no (%)                 0.495
Upper third 5 (25.0) 12 (24.0)
Middle third 6 (30.0) 9 (18.0)
Lower third 9 (45.0) 29 (58.0)

What is the meaning of 0.495 ? I think the tables could be prepared and described in a form easier to understand for those less familiar with statistical methods readers

Since the MUC1 is intended to be used as a prognostic marker can the Authors associate it with the survival of patients?

Author Response

(The authors gave the same response as above.)

Reviewer 3 Report

In this study, the authors found a very good correlation between the malignancy of gastric cancer and the expression level of MUC1 from the results of MUC1-stained biopsies of gastric cancer patients in the United States. This result was astonishing. There are many reports that mention the correlation between the expression level of MUC1 and the malignancy of cancer, but there are many contradictory points. I would like to wait for the results of future large-scale research to see if the cause can be clarified by racial differences. It is epoch-making that MUC1 staining of US gastric cancer early stage biopsy samples enables determination of malignancy and prediction of prognosis. However, there is no benefit to the patient if it cannot be applied to the therapeutic strategy. From the results of GWAS, differences in SNPs in the MUC1 gene depending on the type of gastric cancer have also been clarified. In the future, if we can find differences in sensitivity to anticancer drugs between gastric cancers with high MUC1 expression and those with low MUC1 expression using oncopanels, etc., the importance of this discovery will increase.

Minor Comments:

1. The description of the tissue staining method is ambiguous. The name of the antibody used, including the experimental conditions, should be accurately described so that other researchers can reproduce it in their own laboratory.

2. It would be helpful if you indicated which of A to F in Figure 3 applies during the discussion on lines 140-144 of Section 3.3.

3. The 28.3% listed on line 155 is incorrect. Please correct it to 28.6%.

Author Response

(The authors gave the same response as above.)

Round 2

Reviewer 1 Report

To the authors:

This paper is somewhat well-written original article concerning MUC1 in gastric cancer. In this article, authors revealed that MUC1 expression might be a useful prognostic marker in US GC patientsThe objective of this study is curious, however, there are several points that remain unclear. 

Major Comments;

ESD does not reveal the presence of lymph node metastasis. Also, Stage IV cases are CY-positive and not R0 resected. It may be meaningless to analyze such cases together.

The Stage IV cases is quite high (32%), are they consecutive cases?

Author Response

Thank you for the comments. Our responses are attached as a separate file. Please find the attached file.  
